# Chemical Burn-Induced Corrosive Epiglottitis in an Elderly Patient with Major Depression

**DOI:** 10.3390/life13030804

**Published:** 2023-03-16

**Authors:** Ang Lu, Cheng-Ming Hsu, Yao-Te Tsai, Ming-Shao Tsai, Geng-He Chang

**Affiliations:** 1Department of Otolaryngology-Head and Neck Surgery, Chang Gung Memorial Hospital, Chiayi 613, Taiwan; 2Graduate Institute of Clinical Medical Sciences, College of Medicine, Chang Gung University, Taoyuan 33302, Taiwan; 3Faculty of Medicine, College of Medicine, Chang Gung University, Taoyuan 33302, Taiwan; 4Head and Neck Infection Treatment Center, Chang Gung Memorial Hospital, Chiayi 613, Taiwan

**Keywords:** corrosive, epiglottitis, supraglottitis, psychiatric, depression

## Abstract

Acute epiglottitis (AE) is a potential emergency of the respiratory tract caused mainly by bacterial infection. However, nonbacterial infection causes, such as corrosive injuries, may result in death due to gastrointestinal perforation if a timely diagnosis is not available. We report the case of an elderly patient with an acute melancholic episode who encountered corrosive epiglottitis (CE) caused by accidental ingestion of hydrochloric acid and compare the features of CE and AE, including the immediate onset of symptoms, normal findings on blood tests, and endoscopy revealing pale swollen epiglottitis. This case can prove to be an important reference for clinicians for differential diagnosis, especially when treating epiglottitis in patients with psychiatric disorders and unclear expression.

## 1. Introduction

Acute epiglottitis (AE) is an emergent disease that has an upper airway obstruction risk due to epiglottis swelling and requires high caution and aggressive treatment. Fortunately, the overall incidence of acute epiglottitis has significantly decreased since the conjugated Haemophiles influenzae type B (Hib) vaccine’s subsequent widespread use [1]. The incidence of AE in adults ranges from 1 to 3 per 100,000, and the mortality rate is about 7%. Odynophagia, dysphagia, and voice change are common signs of epiglottitis in adults but are less frequent in children. The common presenting symptoms in children are difficulty breathing and stridor [2]. Besides, more subtle signs and symptoms, such as subjective shortness of breath, tachypnea, and tachycardia, are indicative of a more serious clinical course [1]. Bacterial infection is the primary cause of AE and the Hemophilus influenzae type B is the reported major pathogen [3]. However, in addition to infectious causes, noninfectious factors, such as foreign objects, or associations with systemic disease or reactions to chemotherapy, may also cause acute inflammation and epiglottis swelling [2], including corrosive injury by chemical agents, regarded as corrosive epiglottitis (CE). If CE is initially ignored, it may be complicated by gastrointestinal (GI) hemorrhage, perforation, and even death [4]. However, although CE has been mentioned in the past as a potentially dangerous reason that should be considered when dealing with AE [4], there has been no literature indicating patients who might be more at risk, and no laryngoscopy pictures have been reported that show the appearance of epiglottitis burned by caustic agents. Therefore, we report a case of CE caused by accidental ingestion of hydrochloric acid in an elderly patient with major depression history.

## 2. Case Report

An 88-year-old woman with a history of major depressive disorder and end-stage renal disease undergoing peritoneal dialysis was brought to the emergency room by her family due to general weakness and lying on the lavatory ground. Her chief complaint was sore throat and swallowing pain. In addition to the above symptoms, the patient did not complain of other discomforts such as abdominal pain or fever and she did not mention any chemical ingestion. Physical examination showed that overall appearance, oral cavity, and oropharynx were normal, but the breathing rate was high at 22 per minute, and the oxygen concentration was about 90%. A lateral soft tissue radiograph of the neck was performed and showed a “thumb sign”, suggesting a severely inflamed, edematous, and enlarged epiglottis. However, laboratory studies revealed no leukocytosis and a normal C-reactive protein (CRP) level (1.22 mg/L; normal range: <10 mg/L). Flexible fiberoptic laryngoscopy displayed a swollen but pale epiglottis (Figure 1), causing narrow patency of the airway, consistent with the image of the lateral soft tissue radiograph of the neck.

Based on clinical symptoms and the result of the laryngoscopy, the patient was admitted with a diagnosis of AE, and empiric antibiotic therapy with Ceftriaxone was administered. After admission, she appeared pale in the face the next day and had cold sweating, and fever. Her breathing became more rapid, approximately 24 per minute. Since the symptoms were not improved after antibiotic treatment, we performed repeated laryngoscopy, and it showed that the degree of epiglottis swelling worsened. Blood laboratory tests found that the hemoglobin level decreased from 12.3 mg/dL (normal range: 13.5–17.5 mg/dL for adult males and 12.0–15.5 for adult females) before admission to 8.2 mg/dL. Chest X-ray showed consolidation and pleural effusion of the left lung. On the third day after being hospitalized, her family told doctors that the lid of the hydrochloric acid in their bathroom had been opened, and they suspected that the patient might have accidentally ingested hydrochloric acid because of a depressive episode. Therefore, an urgent gastroscopy was performed and revealed extensive mucosal corrosive injury with multiple hemorrhagic spots from the esophagus to the entire stomach and duodenum (Figure 2). The upper gastrointestinal major bleeding was not seen via gastroscopy. Besides, the ultrasound-guided aspiration of the pleural effusion was performed, and the cytology of pleural contents was transudate effusion. At this point, we changed the diagnosis from bacterial epiglottitis to corrosive epiglottitis with the ingestion of chemical substances. Therefore, we also focused on the treatment of upper gastrointestinal damage. Unfortunately, although we informed the patient and her family about the high mortality rate of this disease progression and recommended to stay in the hospital for ongoing treatment, they refused aggressive treatment, including endotracheal intubation, and further management. Thus, she was discharged from the hospital after signing the informed consent.

## 3. Discussion

Acute epiglottitis is mostly caused by infection, although caustic ingestion, thermal injury, and local trauma are important noninfectious etiologies. AE can occur at any age and may induce upper airway obstruction due to epiglottis swelling, and the disease can progress rapidly and even cause death [2], so it is important to diagnose correctly as early as possible and administer the appropriate treatment. Clinically, the diagnosis of AE is based on patient symptoms, including progressive sore throat, swallowing pain, and imaging studies, such as lateral neck X-ray or endoscopy showing swollen epiglottitis. In addition to the above, the patient’s medical history is also very important. Occasionally, children will drink too hot a liquid, such as hot tea, causing acute swelling of the epiglottis and upper airway obstruction. Since they cannot clearly express their symptoms, doctors must observe other signs to diagnose acute epiglottitis as early as possible. For example, if burns are found on the children’s face, upper thorax, and arms, this type of scalding injury is known as ‘teapot syndrome’, where the hot liquid is grabbed by the children with the aim of ingestion and falls over the children, and then oral, oropharyngeal, laryngeal, and hypopharyngeal damage should be expected. Pediatricians must be aware of acute epiglottitis in order to diagnose and treat it as early as possible [5]. Unlike children, most adults can clearly describe their symptoms to aid in accurate diagnosis. However, in some cases, such as patients with a history of mental illness or elderly people with unclear expressions, they are unable or unwilling to provide their true medical history and it may cause misdiagnosis or delay treatment [4].

Patients with a history of mental illness have a higher chance of intentionally or accidentally ingesting corrosive substances than the general population. Common substances that can cause corrosive injury to the pharynx and larynx include alkaline or acidic substances, such as strong alkalis (sodium hydroxide or potassium hydroxide), strong acids, certain household cleaning products, and batteries [6]. A national population-based study has indicated that patients with psychiatric disorders are more likely to have GI corrosive injury than persons without mental illness [6]. In addition, previous studies have reported that patients with psychiatric disorders have a worse prognosis after accidentally ingesting corrosive substances compared with people without mental illness, which may be related to the higher intake amount [7]. However, some articles had a different point of view and suggested that patients with a history of psychopathology might have a better outcome of corrosive esophageal injury because caregivers are more likely to be on the lookout for warning signs in these patients, and act promptly by bringing them to medical attention. However, in individuals without previous mental illness, suicidal actions may not have been properly discerned, and they therefore present much later for help [8]. Once ingested, corrosive substances, in addition to the damage to the digestive tract, may also cause edema of the epiglottis, which induces upper airway obstruction [4]. In the chronic stages, corrosive ingestion may result in esophageal stricture and increase the risk of upper gastrointestinal cancers [4]. However, if the patient presents only symptoms of sore throat and conceals the intake of corrosive substances, clinicians may focus on upper airway problems while ignoring the damage to the gastrointestinal tract, so the injury of the upper gastrointestinal tract cannot be detected early, delaying necessary tests such as endoscopy, and the timing of treatment. Therefore, clinicians should pay attention to the possibility of CE to facilitate immediate treatment when diagnosing AE in patients with psychiatric disorders and unclear expressions.

AE is mainly caused by bacterial infection, and in addition to the typical symptoms including sore throat and difficulty swallowing, laboratory tests usually show infection characteristics, such as leukocytosis and abnormally elevated CRP levels. Besides, CRP level was found to be a good predictor for airway compromise requiring either intensive care unit (ICU) admission or advanced airway intervention. A threshold CRP level of 100 is set, and once above this threshold, about 75% of patients require protective intubation [9]. These parameters are known to be elevated during acute phase response. However, CE is caused by accidental ingestion of strong acids or alkalis. Therefore, sore throat and odynophagia will occur immediately, and swollen epiglottis development may be more rapid than laboratory tests can show signs of systemic infection. Blood tests may initially show normal white blood cell counts (WBCs, normal range: 4000–11,000/mm^3^) and CRP levels in a patient with CE. Normal laboratory tests may make us underestimate the severity of the disease and delay the timing of treatment. This also reminds us that with patients whose symptoms do not match the laboratory tests, we must be cautious and include CE in the diagnosis.

Although both infectious and noninfectious epiglottitis are characterized by swelling of the epiglottis, it could be found that the swollen epiglottis in CE was significantly different from that of infectious AE in appearance according to the results of the endoscopic examination in this case. Infectious AE usually presents with fiery red and swollen epiglottitis, but the CE case here presented with pale and swollen epiglottitis (Figure 1). The difference in laryngoscopy images between infectious AE and CE is obvious. Although inquiry of medical history is an important process of CE diagnosis, it is difficult to obtain a correct medical history in patients with unclear expressions. Thus, if laryngoscopy and laboratory tests showed a different presentation from infectious AE, such as pale and swollen epiglottitis with an initial normal CRP level, regardless of whether the patient indicates ingestion of corrosive substances, CE must be included in the differential diagnosis. It is important to order appropriate management, such as panendoscopy, to assess for gastrointestinal injury or bleeding.

The clinical presentation of CE depends on the type, amount, and physical form of the substances. The liquid form produces the most caustic effect on the esophagus due to its rapid passage through the mouth and pharynx. In contrast, solid alkali adheres to the mouth and pharynx, causing the most damage to these areas while relatively protecting the esophagus [10]. The main manifestations of GI injury caused by ingestion of corrosive substances are mucosal coagulated necrosis and GI perforation, instead of massive GI hemorrhage. This might be due to necrosis of the entire gastric layer, which turns into a tar-like substance, and to thrombosis of the surrounding blood vessels, causing hemostasis [11]. The extent of damage also depends on several factors, such as the type and the amount of the substance. In the acute stage, GI perforation and necrosis may occur, and long-term complications include glottic and subglottic synechiae and stenosis, esophageal stricture, antral stenosis, and the development of esophageal carcinoma. Short-term stenting for preventing glottic and subglottic synechiae and stenosis is a debatable option for such cases. Endoscopy should be attempted and can be safely performed to assess the extent of damage and give appropriate treatment options such as corticosteroids to prevent stricture formation [10]. Regarding the use of antibiotics, the data are not very clear. However, the consensus is that patients treated with steroids should be treated with prophylactic antibiotics as well [12]. Although esophageal stricture and stenosis are at a considerably lower incidence after corrosive ingestion, health workers should use preventive measures for patients to preclude esophageal strictures during the lengths of hospitalization [6]. The esophageal stent may be placed when esophageal stricture formation following corrosive ingestion does not respond satisfactorily to dilation measures, such as balloon dilation [13]. However, the clinical efficacy of esophageal stent for the relief of dysphagia is approximately 30% and may not last long-term, so placing the stent is considered the last resort for treating esophageal stricture [12]. Some investigators thought that the nasogastric tube, gastrostomy or jejunostomy tube, or central venous total parental nutrition may prevent GI perforation or stricture formation, but the available data are not convincing [12,14]. Nonetheless, esophageal stricture may lead to reflux esophagitis, which may aggravate refractory stricture. Thus, aggressive anti-acid therapy is necessary for patients with corrosive ingestion [6]. In addition to upper gastrointestinal problems, patients with corrosive injuries might also suffer from low airway complications. The complication rate of respiratory failure is about 6.0%, while aspiration pneumonia accounts for 4%. Besides, the previous study also found that low airway and upper gastrointestinal complications were strongly correlated with in-hospital mortality [6].

Antibiotic therapy is necessary for AE, and it is usually initiated without preceding bacterial culture. The first-line antibiotic used is Cefuroxime, followed by Cefuroxime in combination with other antibiotics, such as Metronidazole or Clindamycin [9]. Patients with signs of advancing upper airway obstruction should be treated as an airway emergency. In the presence of respiratory distress, securing the airway should be prioritized over diagnostic procedures and radiography. Although a patient with elevated CRP may have a higher chance of performing tracheal intubation [9], this is not an absolute guideline. Tracheal intubation performance depends on the severity of the respiratory distress. For example, patients with CE may have normal CRP levels at the initial phase, but severe respiratory distress is still possible. Besides, tracheal intubation for a patient with acute epiglottitis is a difficult procedure due to the swollen supraglottic area. Therefore, a team capable of performing an urgent tracheotomy should be verified. After securing the airway, the patient should be transferred to the ICU and intravenous sedation should be used to allow spontaneous ventilation. Dexamethasone therapy may be considered to limit supraglottic edema, thereby reducing the upper airway obstruction [2]. It is reported that the use of corticosteroids has been associated with shorter ICU stays and overall length of admission [1].

Unlike patients with infectious AEs that only need to consider their upper respiratory problems, we should pay more attention to not only upper airway obstruction but also GI injury and systemic complications in patients with CE due to their complicated disease progression. Thus, clinicians should notice latent CE and take appropriate measures when diagnosing and treating epiglottitis to avoid complications or death from the differences in the clinical symptoms appearing over time, as well as the differences between the blood tests and endoscopic findings. This type of epiglottis injury should be suspected, especially in patients with psychiatric disorders or communication difficulties [4]. Furthermore, if the airway and vital signs of patients with CE are stabilized, a psychiatrist should be consulted during the treatment process to evaluate patients’ psychological condition and prescribe adequate psychologic therapy. A close psychiatrist follow-up is needed as well to reduce the chance of chemical suicides.

## 4. Conclusions

Acute epiglottitis is an urgent and life-threatening disease, and the diagnosis mainly depends on the symptoms and swollen epiglottis, seen by flexible fiberoptic laryngoscopy. Careful attention is required in diagnosing and treating epiglottitis, especially in patients with mental disorders and unclear expression. In addition to upper respiratory tract injury, CE injuries can be complicated by GI perforation and even death. Factors such as the onset of CE symptoms, possibly normal WBC and CRP levels, and pale and swollen epiglottitis on laryngoscopy can provide clinicians with an important basis for differential diagnosis.

## Figures and Tables

**Figure 1 life-13-00804-f001:**
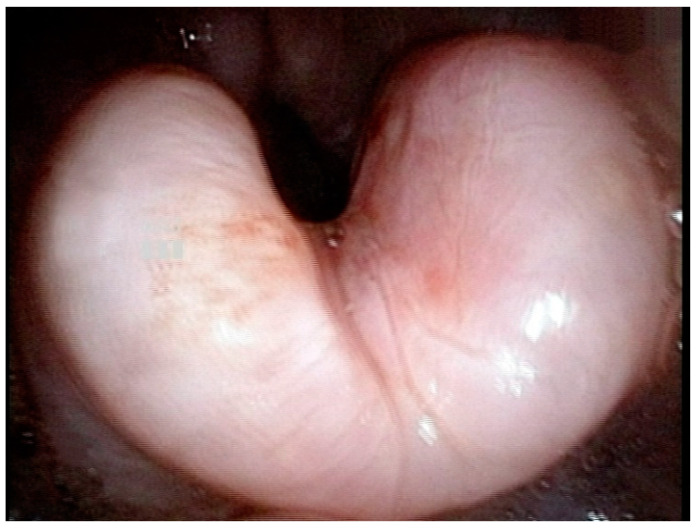
Laryngoscopy revealed a pale swollen epiglottis.

**Figure 2 life-13-00804-f002:**
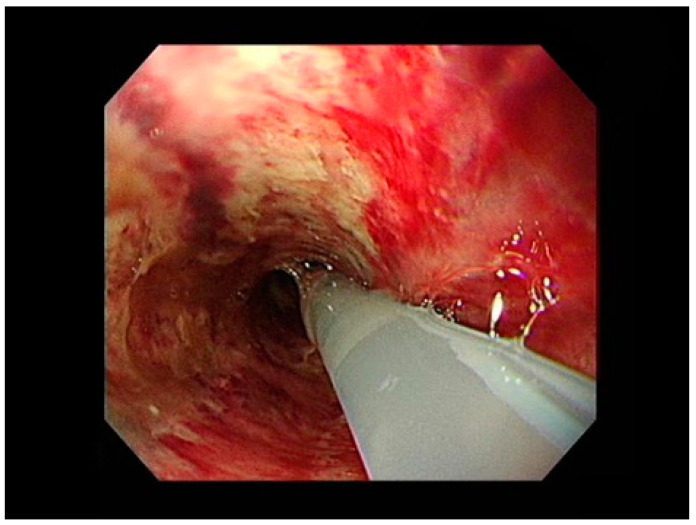
Panendoscopy revealed corrosive mucosal injury with hemorrhage in the esophagus.

## Data Availability

All data underlying the results are available as part of the article and no additional data are required.

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
