# Peer review of "Chemical Burn-Induced Corrosive Epiglottitis in an Elderly Patient with Major Depression"

_life, 2023, doi:10.3390/life13030804_

Round 1

Reviewer 1 Report

I have reviewed this interesting report of a case of corrosive epiglottitis caused by acidental ingestion of hydrochloric acid in a female patient with psychiatric disorder.

Case report: Do the authors know the final outcome of the patient?

Conclusions: Could the last phrasis be extrapollated from the reported case in order to be a conclusion?

Discussion, line 136: please correct “... of the larynx and epiglottis”, since epiglottis is part of the larynx.

Author Response

Comment 1:Case report: Do the authors know the final outcome of the patient?

Reply: Thank you for your questions. Since the patient's family decided to discharge the patient because they refused to allow any further treatment, there is no record of any further emergency or outpatient visits, and we have been unable to contact the patient and family. Therefore, we are unable to determine what the patient's prognosis is.

Comment 2: Conclusions: Could the last phrase be extrapolated from the reported case in order to be a conclusion?

Reply: Thank you for your question. Because the last sentence of the case report is describing the process by which this patient and family were discharged from the hospital because they refused any aggressive treatment, and we were not able to determine the prognosis for this patient, we cannot draw inferences from this text and add a positive conclusion to the conclusions.

Comment 3: Discussion, line 136: please correct “... of the larynx and epiglottis”, since the epiglottis is part of the larynx.

Reply: Thank you very much for your kind reminder, we have corrected the error in the article. (Line: 139)

Thank you again for your time and consideration in reviewing this manuscript.

Reviewer 2 Report

The manuscript provides a clinical case of corrosive epiglottitis, which highlights the importance of history taking in such cases. Solitary elders and patients with poor family support are also common victims of the disease, leading to difficulties in obtaining enough information. Therefore, the manuscript provides valuable experience to raise awareness for initially undiagnosed corrosive epiglottitis. Here are some suggestions to make the article even more helpful:

1. In the discussion section, giving examples of common corrosive injury materials and ideal emergency treatment modalities, in contrast to the outcomes of this case, can be very helpful for clinicians who might encounter such cases. It will help them in providing prompt and appropriate emergency management, leading to better outcomes.

2. Corrosive epiglottitis may also involve the glottis and subglottis area, leading to severe synechiae and stenosis in the long term. Short-term stenting is a debatable option for such cases. A brief discussion can be added to the article for the readers to understand the potential benefits and risks.

Author Response

Comment 1:In the discussion section, giving examples of common corrosive injury materials and ideal emergency treatment modalities, in contrast to the outcomes of this case, can be very helpful for clinicians who might encounter such cases. It will help them in providing prompt and appropriate emergency management, leading to better outcomes.

Reply: Thank you very much for this wise and important advice. We have included a section in the Discussion on common substances that cause corrosive injury and have also revised the text to describe the proper management of such patients in the emergency setting to avoid inadvertent diagnosis and treatment of CE patients. (Line: 124-127; 169-176)

Comment 2: Corrosive epiglottitis may also involve the glottic and subglottic area, leading to severe synechiae and stenosis in the long term. Short-term stenting is a debatable option for such cases. A brief discussion can be added to the article for the readers to understand the potential benefits and risks.

Reply:Thank you very much for reminding us of this important issue. It is true that glottic and subglottic synechiae and stenosis can occur in such patients. The treatment of short-term stent placement in this area is still not clearly recommended. We have made changes in the appropriate paragraphs of the Discussion to reinforce the possible complications of glottic and subglottic areas. (Line: 187-190)

Thank you again for your time and consideration in reviewing this manuscript.

Reviewer 3 Report

This manuscript described a case of corrosive epiglottitis in an elderly patient with major depression who ingested hydrochloric acid. Acute epiglottitis is an urgent, life-threatening disease which could obstruct the airway thus clinicians should be well aware of this disease. The manuscript was well written but there are several minor items to modify.

1. In line 56, might more -> might be more.

2. In lines 149 and 219, present the full terminology and abbreviation of ICU at the first appearance.

3. In line 192, maybe -> may be.

4. In line 200, refluxesophagitis -> reflux esophagitis.

Author Response

Comment 1:

  1. In line 56, might more -> might be more.
  2. In lines 149 and 219, present the full terminology and abbreviation of ICU at the first appearance.
  3. In line 192, maybe -> may be.
  4. In line 200, refluxesophagitis -> reflux esophagitis.

Reply:Thank you very much for your kind reminder. We have fixed all the errors based on your correction. (Line: 56; 152-153; 197; 205)

Thank you again for your time and consideration in reviewing this manuscript.

Reviewer 4 Report

thank you for the opportunity to read an interesting work on diagnostic difficulties in cases of chemical burns of the respiratory tract and esophagus.

Authors should complete the work: The title of the work should contain a statement that it concerns the case of a chemical burn.

Based on the presented material, it can be assumed that we are dealing with a suicide attempt and not accidental consumption of a corrosive agent in a person with depression, therefore, was there a psychiatrist's consultation?

The work should include the range of standards for laboratory tests that are presented.

Author Response

Comment 1:Authors should complete the work: The title of the work should contain a statement that it concerns the case of a chemical burn.

Reply: 

Thank you very much for your insightful advice. We have corrected the title to better represent the content of this case report. (Line: 2)

Original: Corrosive epiglottitis in an elderly patient with major depression

Revised: Chemical burn-induced corrosive epiglottitis in an elderly patient with major depression

Comment 2: Based on the presented material, it can be assumed that we are dealing with a suicide attempt and not accidental consumption of a corrosive agent in a person with depression, therefore, was there a psychiatrist's consultation?

Reply:

Thank you very much for your thoughtful comment. By the time we diagnosed epiglottitis due to corrosive injury, the patient's respiratory and vital signs had already deteriorated, so we focused on treating the patient's acute symptoms. However, because the family did not want to allow the patient to undergo further aggressive treatment such as intubation, and the patient was discharged after signing a consent form, we have not had the opportunity to consult with psychiatry to assess the patient's psychiatric status.

However, we did add a paragraph at the end of the discussion that after the patient's respiratory and vital signs are under control, it is recommended to consult a psychiatrist to assess the patient's psychological status during the treatment process and to follow up and provide appropriate treatment in order to reduce the chance of suicide. (Line: 237-240)

Comment 3:The work should include the range of standards for laboratory tests that are presented.

Reply:Thank you very much for your professional advice. We have included the normal range of values in the article where the test data are presented, including white blood count (WBC), C-reactive protein (CRP) and hemoglobin (Hb) data. (Line: 71; 83-84; 159)

Thank you again for your time and consideration in reviewing this manuscript.

Round 2

Reviewer 4 Report

accept in present form